# Killing by Degradation: Regulation of Apoptosis by the Ubiquitin-Proteasome-System

**DOI:** 10.3390/cells10123465

**Published:** 2021-12-08

**Authors:** Ruqaia Abbas, Sarit Larisch

**Affiliations:** Laboratory of Cell Death and Cancer Research, Biology & Human Biology Departments, Faculty of Natural Sciences, University of Haifa, Haifa 3498838, Israel; rabbas18@campus.haifa.ac.il

**Keywords:** ubiquitin proteasome system, ARTS, Smac, XIAP, cIAP, Bcl-2, Mcl-1, parkin, p53, MDM2

## Abstract

Apoptosis is a cell suicide process that is essential for development, tissue homeostasis and human health. Impaired apoptosis is associated with a variety of human diseases, including neurodegenerative disorders, autoimmunity and cancer. As the levels of pro- and anti-apoptotic proteins can determine the life or death of cells, tight regulation of these proteins is critical. The ubiquitin proteasome system (UPS) is essential for maintaining protein turnover, which can either trigger or inhibit apoptosis. In this review, we will describe the E3 ligases that regulate the levels of pro- and anti-apoptotic proteins and assisting proteins that regulate the levels of these E3 ligases. We will provide examples of apoptotic cell death modulations using the UPS, determined by positive and negative feedback loop reactions. Specifically, we will review how the stability of p53, Bcl-2 family members and IAPs (Inhibitor of Apoptosis proteins) are regulated upon initiation of apoptosis. As increased levels of oncogenes and decreased levels of tumor suppressor proteins can promote tumorigenesis, targeting these pathways offers opportunities to develop novel anti-cancer therapies, which act by recruiting the UPS for the effective and selective killing of cancer cells.

## 1. Apoptosis (Programmed Cell Death)

Apoptosis is a morphologically and mechanistically distinct cell death program that is essential for the elimination of unwanted and damaged cells during development and tissue homeostasis [1,2,3]. Abnormal regulation of this process is associated with a wide variety of human diseases, including immunological and developmental disorders, neurodegeneration and cancer [2,3,4]. Apoptosis is executed by caspases, enzymes that are activated following cleavage from their inactive pro-caspase form [5]. Caspase-dependent cell death is the hallmark of apoptosis. However, several alternative modes of non-apoptotic cell death have been described that do not involve caspases [6,7,8,9,10,11]. Apoptosis is regulated by two major pathways: the extrinsic pathway and the intrinsic (mitochondrial) pathway [1,12]. The extrinsic pathway is initiated when apoptotic-inducing ligands bind to death receptors [13]. The intrinsic pathway is mainly induced by internal apoptotic signals, such as DNA damage, as well as by certain external stimuli, such as nerve growth factor (NGF) withdrawal-induced cell death [14]. There is an essential crosstalk between the extrinsic and intrinsic pathways, for example, via caspase-induced-cleavage of BID. Truncated Bid (tBID) is initially cleaved in the extrinsic pathway by caspase-8 and can amplify the apoptotic signal by translocating to the mitochondria and inducing mitochondrial outer membrane permeabilization (MOMP) [15,16,17]. This in turn leads to the release of Cytochrome c and formation of the apoptosome complex, which activates caspase-9 and, subsequently, downstream effector caspases [5,16,17,18,19]. This initiates a proteolytic cascade, which culminates in the cleavage of substrate proteins, leading to the disassembly of the cell [17]. In living cells, caspases are kept controlled by inhibitors of apoptosis (IAP) proteins. The IAP proteins prevent cell death by binding to and inhibiting active caspases [20]. XIAP (X-linked IAP), is the best studied IAP, which has three Baculovirus IAP Repeat (BIR) domains. XIAP-BIR3 domain binds directly to and inhibits caspases -9, and the linker region between XIAP-BIR1 and BIR2 domains inhibits caspase-3 and -7 [21,22,23]. In contrast, cIAP1 and cIAP2 can bind to but not inhibit caspases [24,25]. Both XIAP and cIAPs contain a RING (Really Interesting New Gene) domain, which enables these proteins to function as E3 ligases [26,27,28,29] (see also IAPs section below). Apoptosis is a highly regulated process with pivotal checkpoints. Cancer cells evade apoptosis by disrupting these checkpoints [30,31,32]. The ubiquitin proteasome system (UPS) plays a critical role in keeping this fine-tuned process in check by tightly regulating the levels of anti- and pro-apoptotic proteins [19,33,34,35,36,37,38,39,40,41,42].

## 2. The Ubiquitin Proteasome System

The Ubiquitin Proteasome System (UPS) is responsible for the regulated degradation of intracellular proteins [43,44,45]. Ubiquitylation is the post-translational conjugation of the 76 amino acid ubiquitin protein, which tags proteins destined for degradation via the 26S proteasome [45,46,47,48,49,50]. Ubiquitylation modulates many cellular processes, including DNA replication, DNA repair, cell viability, transcription and apoptosis [51,52,53,54,55,56,57,58,59,60]. The ubiquitin molecule can form eight different polyubiquitin chains by employing one of its seven lysine residues (K6, K11, K27, K33, K48 or K63) or by the methionine residue at position 1 (M1) [61]. Protein modification can be in the form of a mono-, multi-mono- or poly-ubiquitin chains [58,61,62,63,64,65,66,67]. The K48 and the K63 chains account for 80% of the total linkages in mammalians. The K48 chains are involved in proteasome-dependent degradation [68]. In addition, K11 and K29 chains were also found to promote proteasome-dependent degradation [61,69]. In contrast, the levels of K63 chains do not change during proteasome inhibition [70,71,72,73,74,75]. K63 promotes proteasome-independent pathways, such as inflammatory signal transduction, autophagy, endocytosis and DNA repair [76,77]. The ubiquitylation cascade requires a ubiquitin activating enzyme (E1), ubiquitin conjugating enzymes (E2) and the ubiquitin ligases (E3) [78]. The E1 enzyme forms an ATP-dependent thioester linkage with the carboxyl-terminus of ubiquitin; E1 is not substrate-specific [79]. The E2 enzyme receives the activated ubiquitin from E1, which in turn transfers the ubiquitin to the E3 ligase [80,81]. Different E2 enzymes can regulate a single E3 ligase [80,81]. The E3 ligases are substrate-specific and are essential for the final transfer of the activated ubiquitin from the E2 enzyme to the lysine residue onto the target protein [80,81,82].

### E3 Ligases

E3 ligases are the largest and most studied group of the UPS; they can be classified into three major groups. The RING E3 ligase family is the largest, containing 600 members. The HECT family, homologous to Human Papilloma virus E6 Carboxyl Terminal domain, contains 28 members and the RBR (RING between RING fingers) E3 ligase family contains 18 members [83,84,85,86,87]. Many studies have shown that E3 ligases play an important role in oncogenesis by mediating chemo-resistance [88,89,90,91,92,93,94,95,96,97,98,99,100,101]. Hence, any disruption in regulating protein homeostasis via the UPS will result in carcinogenesis and chemo-resistance [102,103,104,105,106,107,108].

The RING E3 ligase family are characterized by the presence of a zinc-binding domain called RING or by a U-box domain [19,109]. The U-box domain is structurally similar to the RING domain but does not contain zinc [110]. The RING E3 ligases do not bind to the ubiquitin itself but rather bind the E2 enzymes carrying activated ubiquitin via the Zn^+2^ within the RING domain [19]. RING E3 ligases can function as monomers, homodimers or heterodimers [111]. The U-box domains can work as either monomers or homodimers [111]. Some RING E3 ligases consist of multiple subunits, such as cullin-RING ligase. Cullins are a family of scaffold proteins that have crucial roles in post-translational modification of proteins involving ubiquitin [112,113]. The cullin RING ligases are composed of cullin as a scaffold proteins that binds a RING-box at its N-terminus, an adaptor protein and a substrate receptor at its C-terminus [112].

The HECT family of E3 ligases acts as an intermediate between the E2 enzyme and target proteins. In these E3 ligases, the ubiquitin is transferred to the cysteine residue of the HECT domain, which is then transferred to the target proteins [84,85].

The RBR E3 ligases are known to have two RING domains (RING1 and RING2) that are linked by an “In-between RING” (IBR) domain [114,115]. Similar to HECT E3 ligases, the RBR E3s act as an intermediate between the E2 enzymes and their substrates [19]. The RING1 binds to the ubiquitin-charged E2 enzyme and allows the transfer of ubiquitin from the E2 enzyme to the cysteine residue of RING2, which in turn transfers it to the target protein [19,114,115].

## 3. The Regulation of Pro- and Anti-Apoptotic Protein by the UPS

The Ubiquitin Proteasome system (UPS) regulates apoptosis by targeting proteins, which are important for executing the apoptotic process. Modulation of apoptosis by the UPS can either initiate or inhibit apoptosis [44,116,117]. The role of ubiquitin in cell death was first reported for the non-apoptotic death of muscle cells during insect metamorphosis [118]. As insects transition from the larval to the pupal stage, the steroid hormone ecdysone triggers the elimination of many larval tissues [119]. The degradation of larval intersegmental muscles was one of the first descriptions of developmental programmed cell death [120]. Ironically, the exact mechanism underlying larval muscle histolysis is still unclear. In this review, we will focus on the regulation of apoptotic proteins by the UPS.

### 3.1. p53

*p53* is the most intensively studied gene in molecular oncology [121,122]. Nearly 50% of all human tumors have mutations in the p53 gene [122,123,124]. In some cases, the mutated p53 protein is stabilized by evading proteasome-dependent degradation [125]. Since abnormalities in p53 levels can cause tumorigenesis, UPS-mediated degradation is important to keep the levels of p53 in check [126,127,128,129,130,131]. p53 is a major transcription factor that controls a variety of cellular processes, such as growth arrest, DNA repair, senescence and apoptosis [132,133]. In viable cells, p53 is present at low levels and has a short half-life of approximately 5 to 20 min [134,135,136,137,138]. However, following stress signals, including DNA damage, the half-life of p53 increases several-fold through inhibition of its E3 ligases, causing p53 protein accumulation [136,137,138,139,140]. So far, 20 E3 ligases that regulate p53 levels via the UPS have been identified (Table 1, Figure 1). The onco-protein MDM2 is the main E3 ligase that mediates p53 ubiquitylation [141,142,143]. The N-terminal part of MDM2 obstructs the transcription function of p53, while its C-terminal part recruits the E2 ubiquitin conjugate enzyme for ubiquitylation and degradation of p53 [144,145,146]. *MDM2* knockout (KO) mice are embryonically lethal due to excessive p53 accumulation [147,148]. This phenotype can be rescued by the inactivation of p53 [147,148]. Notably, p53 and MDM2 operate in a negative feedback loop (Figure 2); p53 induces the expression of MDM2, which in turn promotes the degradation of p53 [133,149,150,151]. Many tumors display high levels of MDM2 and low levels of p53 and thereby manage to evade apoptosis [144].

Regulation of apoptosis by the UPS also involves other proteins that modulate E3 ligase activity and thereby control the levels of pro-apoptotic proteins. MDM2 is an unstable protein that undergoes ubiquitylation and degradation in an autocatalytic manner [152,153]. MDM2 exhibits increased E3 ubiquitin activity and self-ubiquitylation when it forms homo- or hetero-dimers [154,155]. Moreover, MDM2 stability increases when certain proteins bind to its C-terminal RING domain [156,157,158]. For example, MDM2 activity is regulated by MDMX, a protein that shares sequence similarity with MDM2 but lacks the E3 ligase activity [159]. MDMX and MDM2 form heterodimers that enhance the MDM2 E3 ligase activity and reduce MDM2 auto-ubiquitylation and degradation [158,160,161]. MDMX is essential for the suppression of p53 by MDM2, as MDMX KO mice, similar to MDM2 KO mice, are embryonically lethal due to increased p53 activity. Similar to the MDM2 KO phenotype, these mice can be rescued by the co-deletion of p53 [162,163,164]. In addition, the MDM2 RING domain binds the internal ribosome entry site (IRES) region in XIAP mRNA, thereby preventing homodimerization of MDM2 and its autoubiquitylation. This results in increased MDM2 stability and XIAP translation, leading to cell survival [157,165]. Moreover, p53 promotes the expression of the ubiquitin domain-containing 1 (UBTD1) protein, which in turn induces MDM2 degradation through a positive feedback loop mechanism [166]. Although MDM2 is considered the primary E3 ligase of p53, other E3 ligases were shown to control the stability of p53. Some E3 ligases can promote the degradation of p53 independently of MDM2, such as Pirh2, COP1, ARF-BP1, CHIP, TOPORS, Synoviolin and Carps (see Table 1 for a complete list). Three additional E3 ligases, TRIM28, RNF2 and Cul4a, can promote p53 degradation by interacting with MDM2.

p53-induced RING-H2 protein (Pirh2) is an important transcriptional target of p53. Pirh2 can physically interact with p53, inducing its ubiquitylation and degradation independently of MDM2, which initiates a negative feedback loop (Figure 2) [167,168,169]. Various kinds of cancers express high levels of Pirh2, which is associated with poor prognosis and survival rate [169,170,171]. Using hepatocellular carcinoma samples, the SCYL1-binding protein 1 (SCYL1BP1) was found to promote Pirh2 degradation and thereby restore p53 levels [172]. The COP1 protein contains an N-terminal RING finger and can ubiquitylate p53 independently of MDM2 and Pirh2 under stressed and unstressed conditions [173]. However, COP1^hypo/−^ mice do not exhibit any dysregulation in p53 levels, suggesting that COP1 does not play a role in regulating the levels of p53 [174]. ARF-BP1 (HUWE1, MULE) is a HECT E3 ligase that can also regulate p53 levels independently of MDM2 [175]. Silencing of the ARF-BP1 expression by RNAi in U2OS cells resulted in p53-dependent apoptosis [175]. The TRIM protein family has an N-terminal RING finger domain and at least one B-box zinc finger domain [176]. TRIM28 can induce p53 ubiquitylation and degradation via its interaction with MDM2 [177]. Yet, silencing TRIM28 caused an increase in the expression of p53 target genes [177,178]. U-box E3 ligases can also induce p53 ubiquitylation and degradation. On example is CHIP (carboxyl terminus of Hsc70-interacting protein), a chaperone-interacting protein that has E3 ligase activity and is responsible for the ubiquitylation and degradation of various proteins via its C-terminal U-box [179,180]. CHIP can promote p53 degradation by interacting with Hsc70 [179,180]. CHIP ubiquitylates and degrades both wild-type and mutant p53 via both the proteasome and the lysosome pathways [181,182]. Since p53 regulates various cellular processes, it has been suggested that each E3 ligase is assigned for regulating p53 under certain conditions or in specific tissues or cell types [183].

p53 promotes apoptosis mainly by inducing the transcriptional upregulation of pro-apoptotic proteins, such as the death receptor 5 (DR5), TNFR1 and Fas, which results in the activation of caspase-8 [184,185]. In addition, p53 can induce the activation of pro-apoptotic Bcl-2 family proteins, such as BAX, PUMA, BAD, BID, BAK and NOXA [186,187,188,189,190]. The induction of BID, BAK and BAX promotes the permeabilization of the outer mitochondrial membrane and amplifies the caspase activation process [191,192]. p53 also increases the transcription of the pro-apoptotic XIAP-antagonist, ARTS, which relieves caspases from inhibition by XIAP, leading to the cleavage of BID and MOMP [190,193].

**Table 1 cells-10-03465-t001:** Ubiquitin E3 ligases targeting p53 for degradation.

E3 Ligase	Type	Effect on p53	Model	Ubiquitination Observed In Vivo/In Vitro	References
MDM2	RING	Degradation	mouse	In vivo and in vitro	[141,142]
ARF-BP1	HECT	Degradation	mouse	In vivo and in vitro	[175]
CHIP	U-box	Degradation	mouse	In vivo and in vitro	[182]
Cop1	RING	Degradation	-	In vivo and in vitro	[173]
Cul1	RING	Degradation	mouse	In vivo and in vitro	[194]
Cul4a	RING	Degradation	mouse	In vivo	[195,196,197]
Cul5	RING	Degradation	-	In vitro	[198]
Synoviolin	RING	Degradation	drosophila	In vivo and in vitro	[199]
TOPORS	RING	Degradation	-	In vivo and in vitro	[200]
Trim24	RING	Degradation	drosophila	In vivo and in vitro	[201]
TRIM28	RING	Degradation	-	In vitro	[202]
TRIM39	RING	Degradation	-	In vitro	[203]
TRIM65	RING	Degradation	-	In vivo	[204]
Carpi	RING	Degradation	-	In vivo	[205]
Carp2	RING	Degradation	-	In vivo	[205]
Pirh2	RING	Degradation	mouse	In vivo and in vitro	[169]
TRAF6	RING	Degradation	mouse	In vivo and in vitro	[206]
TRAF7	RING	Degradation	-	In vitro	[207]
RNF2	RING	Degradation	mouse	In vivo and in vitro	[208]
RING1	RING	Degradation	-	In vivo and in vitro	[209]

### 3.2. BCl-2 Family

Members of the Bcl-2 family of proteins mainly modulate the intrinsic (mitochondrial) pathway [210,211]. This family contains both pro- and anti-apoptotic proteins that share up to four conserved Bcl-2 homology (BH) domains and form complexes by binding to their common BH3 domains [212,213]. The balance between pro- and anti-apoptotic proteins of the Bcl-2 family determines the sensitivity of cells to apoptotic stimuli [214]. The Bcl-2 family can be divided into three subgroups: anti-apoptotic (Bcl-2, Bcl-xL, Bcl-w, Mcl-1, A1 and Bcl-B), pro-apoptotic (BAX and BAK), and pro-apoptotic BH3-only proteins (Bim, Bad, tBid, Bmf, Bik, Noxa, Puma and Hrk) [211,214]. Bcl-2 itself is a key cell death regulator that inhibits cell death [215]. Many cancers are characterized by high levels of Bcl-2 [212,213,216,217,218]. In apoptotic cells, Bcl-2 levels decrease as a result of their degradation by the UPS [36,219,220,221]. XIAP serves as the E3 ligase for Bcl-2 (Figure 1) [36]. Upon apoptotic stimuli, Bcl-2 is brought into close proximity to XIAP by ARTS, which serves as a scaffold protein. The formation of a ternary complex between Bcl-2, ARTS and XIAP enables Bcl-2 ubiquitylation by XIAP and its subsequent degradation by the proteasome [36].

The myeloid cell leukemia 1 (Mcl-1) protein is involved in attenuating the apoptotic response upon DNA damage, growth factor withdrawal and adenoviral infection [222]. Proteasome-mediated degradation of Mcl-1 is crucial for its proper function [223,224]. The first identified E3 ligase of Mcl-1 was the “Mcl-1 ubiquitin E3 ligase” (MULE/ARF-BP1). MULE contains a BH3 domain that assists in binding to Mcl-1 but not to Bcl-2 or Bcl-xL [222,225]. Another E3 ligase that regulates Mcl-1 during neuronal apoptosis is Tripartite Motif-containing protein 17 (TRIM17) [226]. Parkin is yet another E3 ligase that is involved in the degradation of Mcl-1 during mitochondrial depolarization [227]. Furthermore, SCF^β-TrCP^ and SCF^FBW7^ are two additional E3 ligases of Mcl-1 that belong to the SCF family (Skp1, Cullin, F-box complex) [228,229,230]. Finally, APC/C^Cdc20^ (APC/C complexed with substrate recognition adapter Cdc20) was proposed to mediate Mcl-1 ubiquitylation during mitotic arrest (Figure 1) [231,232]. This long list of E3 ligases targeting Mcl-1 illustrates the extent and complexity that is devoted to the Ub-mediated degradation of this protein.

BID is an important pro-apoptotic Bcl-2 family protein. In response to apoptotic stimuli, BID is cleaved to generate tBID, which in turn binds to BAX and BAK and promotes MOMP [5,16,17,18,19]. tBID connects the extrinsic apoptotic signaling pathway with the intrinsic pathway, thus allowing the amplification of external apoptotic stimuli [5,16,17,18,19]. BID levels are also tightly regulated by E3 ligases, such as the Itchy homolog E3 ligase (ITCH/AIP4). ITCH/AIP4 specifically binds to and ubiquitylates tBID but not the uncleaved form of BID (Figure 1) [233].

In living cells, Bax is mainly present in the cytosol. However, in response to apoptotic stimuli, Bax is activated by undergoing a conformational change that causes translocation to the mitochondrial outer membrane (MOM) [234,235,236]. There, BAX binds to BAK to initiate MOMP, allowing the release of pro-apoptotic proteins. These include Smac/Diablo (Smac) and Cytochrome c (Cyto c), which normally reside in the mitochondrial inner membrane space (IMS) [237]. Since activation of Bax and its translocation to the MOM act as a major regulatory checkpoint in apoptosis, Bax protein levels are tightly controlled by the UPS [238,239,240]. The in-between RING (IBR) domain containing 2 (IBRDC2) E3 ligase induces the degradation of BAX in response to p53-mediated apoptosis [241]. IBRDC2 is highly specific for BAX, as it does not bind to BAK, PUMA or NOXA [241]. Johnson et al. identified Parkin as another E3 ligase that can ubiquitylate BAX, thereby limiting the mitochondrial pool of BAX under non-apoptotic and stress conditions (Figure 1) [242].

### 3.3. Inhibitor of Apoptosis (IAPs) Proteins and IAPs-Antagonists

In viable cells, the activity of caspases is inhibited by IAPs [20,24]. The IAP family consists of eight members in mammals: XIAP, cIAP1, cIAP2, ML-IAP, NAIP, ILP2, Survivin and Bruce [243]. IAPs contain between one to three Baculoviral IAP repeat (BIR) domains, which act as a protein–protein interaction domain [5,17,25]. In addition, XIAP, cIAP1, cIAP2, ML-IAP and ILP2 possess a ubiquitin-associated (UBA) domain that assists in the binding of poly-ubiquitin conjugates and a RING domain responsible for their E3 ligase activity [26,27,28,29]. XIAP is the most potent member of the IAP family in terms of its ability to directly inhibit caspases and suppress apoptosis [244]. cIAPs interact with TNF-associated factors (TRAFs) to hinder the formation of pro-apoptotic signaling complexes in the extrinsic apoptotic pathway initiated by TNFR [245,246,247,248]. It is noteworthy that cIAPs mediate cell survival through both canonical and non-canonical NF-κB signaling [247,249,250,251,252,253,254]. Numerous tumors overexpress XIAP and cIAPs, which enables cancer cells to escape apoptosis [255,256]. Accordingly, both XIAP and cIAPs have become promising targets for cancer therapy [257,258,259,260]. In cells undergoing apoptosis, IAP-antagonists inactivate IAPs, leading to de-repression of active caspases [261,262]. The mammalian IAP-antagonist proteins are Smac, Omi/HtrA2, XAF1 (XIAP-associated factor 1) and ARTS [37,257,263,264,265,266,267,268]. Smac and Omi/HtrA2 contain a conserved four-amino acid domain (AVPI/F), which was first described in the *Drosophila* IAP-antagonists Reaper, Hid and Grim and termed IBM (IAP-binding motif) [269,270,271,272]. Genetic and biochemical characterization of *reaper*, *hid*, *grim* and *diap1* (*Drosophila* IAP1) provided the first evidence for the critical physiological role of IAPs and their antagonists in regulating apoptosis [271,273,274,275,276]. Smac resides in the inner-membrane space of mitochondria [263,264,277]. Upon apoptotic induction, Smac and Cytochrome C (Cyto c) are released into the cytosol [263]. Cyto c, together with APAF-1 and pro-caspase-9, then form the “apoptosome” complex, which cleaves and activates caspase-9 [278]. Smac binds to the caspase-9 pocket in the BIR3 domain of XIAP via its IBM, resulting in the release of XIAP-bound-caspases [263,279,280,281]. Smac binds to cIAP1, cIAP2 and XIAP, yet it only induces ubiquitylation and degradation of cIAPs but not XIAP (Figure 2) [282,283]. ARTS (Sept4_i2) is a splice variant derived from the Sept4 (Septin 4) gene and the only splice variant that functions as a pro-apoptotic protein [284]. ARTS is a tumor-suppressor protein that is localized at the Mitochondrial outer membrane (MOM) [193]. Upon apoptotic stimuli, ARTS rapidly translocates to the cytosol in a caspase-independent manner, where it binds and antagonizes XIAP [37,193]. Furthermore, the translocation of ARTS from the MOM to the cytosol precedes MOMP and the release of Cyto c and Smac and is required for it [36,193]. The localization of ARTS at the MOM facilitates its rapid translocation to the cytosol and binding to XIAP within minutes following apoptotic stimuli [193]. The direct binding of ARTS to XIAP enables de-repression of caspases, which are required for MOMP, and the subsequent release of Cyto c and Smac [36,193,279,285,286,287,288,289,290]. Under steady state conditions, XIAP inhibits unwanted apoptosis by promoting the degradation of active caspase-9, -3 and the apoptosis-inducing factor (AIF) through the UPS (Figure 1) [55,291,292,293]. Furthermore, XIAP attenuates apoptosis by degrading two substrates—both its antagonists Smac and ARTS (Figure 1) [34,294,295]. ARTS levels are also regulated by the E3 ligase Parkin (Figure 1) [296]. ARTS acts as the physiological antagonist of XIAP as Sept4/ARTS KO mice exhibit high levels of XIAP [33,35,37,297,298,299]. Consequently, these ARTS-deficient mice develop various spontaneous cancers— mainly lymphoma and leukemia [299]. Importantly, ARTS is also required for the ubiquitylation and degradation of XIAP by another E3 ligase, seven in absentia homolog (SIAH) (Figure 1 and Figure 2) [298]. cIAP1 E3 ligase activity affects the levels of Smac (in a negative feedback manner) (Figure 2), in addition to TRAF2, TRAF3 and FLIP_L_ (Figure 1) [249,293,300,301,302,303,304]. ML-IAP is another E3-ligase that regulates the levels of Smac, and it has been proposed that this may serve to inhibit apoptosis and promote drug resistance in melanoma cells (Figure 1) [29,305]. Finally, the HECT (homologous to E6-AP carboxyl terminus) family E3 ubiquitin ligase AREL1 can inhibit apoptosis by ubiquitylating all three IAP antagonists Smac, HtrA2 and ARTS [306]. Therefore, UPS-mediated protein degradation plays a critical and complex role for both the function and regulation of IAPs and their antagonists.

## 4. Targeting the UPS for Apoptosis-Induced Cancer Therapy

Cancer cells engage the UPS for degrading pro-apoptotic proteins [307,308]. Hence, evading apoptosis by tipping the balance between pro- and anti-apoptotic proteins may allow initiation of tumorigenesis [4,309,310,311,312]. Intensive efforts are being made to use the UPS for killing cancer cells. The original strategy was to develop general proteasome inhibitors for the treatment of multiple myeloma, which lead to several FDA-approved drugs for the treatment of multiple myeloma (see below) [313,314,315]. This success has encouraged additional efforts to develop new cancer therapeutics by targeting UPS-mediated protein degradation [316,317,318].

### 4.1. Proteasome Inhibitors

At this time, three proteasome inhibitors have been approved by the FDA for the treatment of multiple myeloma: Bortezomib (Velcade, PS341), its second-generation derivative Carfilzomib (Kyprolis) and Ixazomib (MLN9708, Ninlaro) [313,314]. Bortezomib was the first proteasome inhibitor to be approved by the FDA in 2003 [319,320,321,322]. Bortezomib inhibits the 20S proteasome subunit, affecting several vital cellular pathways, including the NF-κB signaling, thereby promoting apoptosis [320,323]. Moreover, Bortezomib increases the levels of the pro-apoptotic protein NOXA [324]. Carfilzomib was the second proteasome inhibitor to be approved by the FDA in 2012 [325,326]. Carfilzomib is a more potent inhibitor of the proteasome when compared to Bortezomib, and it is effective against Bortezomib-resistant multiple myeloma [325,326]. Carfilzomib is thought to induce apoptosis by increasing NOXA levels, which results in the activation of capase-3 and -7 [327]. Unfortunately, despite its effectiveness, Carfilzomib shows dose-limiting toxicities [112].

Ixazomib is the first orally administered proteasome inhibitor; it is as effective as Bortezomib, with respect to inhibiting the proteasome, but has better pharmacokinetic properties. Ixazomib was approved by the FDA and is administered in combination with lenalidomide and dexamethasone in patients with relapsed and refractory myeloma [328,329]. Significantly, Ixazomib can induce apoptosis in Bortezomib-resistant multiple myeloma patients [330]. Delanzomib is another orally administered proteasome inhibitor that inhibits NF-κB signaling and can promote apoptosis in multiple myeloma [331]. Delanzomib is also more effective than Bortezomib in treating normal human epithelial bone marrow progenitor and bone marrow-derived stromal cancer cells [331]. Despite its relative effectiveness, phase II clinical trials with Delanzomib were terminated due to considerable toxicity [332]. Another proteasome inhibitor is Oprozomib, an oral tripeptide epoxyketone. Oprozomib has a longer half-life than Bortezomib and causes activation of caspases-9, -3 and -7 and apoptosis [333,334,335]. Finally, Marizomib is the first natural proteasome inhibitor derived from *Salinosporamide tropica*, a marine actinomycete bacterium [336]. Marizomib causes irreversible inhibition of the 20S proteasome when tested in in vitro and in vivo models [336]. Both Oprozomib and Marizomib underwent clinical trials as a single agent or in combination with other drugs [329]. Despite the relatively high efficacy of general proteasome inhibitors to treat multiple myeloma, their long-term use is limited due to aqcuired resistance towards these compounds [337].

### 4.2. Cancer Therapies Targeting p53 and MDM2 for Degradation

Because of their vital role in maintaining protein turnover, E3 ligases are also complicit in assisting tumorigenesis. Therefore, E3 ligases present promising drug targets for cancer treatments. The inhibition of E3 ligases is supposed to be more target-specific and show lower toxicity compared to general proteasome inhibitors [316].

MDM2 is essential for restricting the levels of p53. Various cancers overexpress MDM2 to hinder the p53-mediated pathway, thus resulting in tumor progression [338,339]. Therefore, MDM2 became an emerging target for developing cancer treatments [316]. The Nutlin small molecules are a family of cis-imidazoline analogs first described as selective and potent inhibitors of MDM2 [340,341]. Nutlins can occupy the binding site of p53 in MDM2 and allow p53 to escape MDM2-mediated ubiquitylation and degradation [340]. Amongst the Nutlin family members, only the enantiomer Nutlin-3a exhibited a potent binding ability to MDM2. Yet, these molecules were not effective enough to be further examined in clinical trials [342]. Resolving the crystal structure of Nutlin-3a facilitated the discovery of better MDM2-p53 inhibitors, such as RG7112, which is currently in phase I clinical trials [343,344,345]. Other small molecules that target the interaction between MDM2 and p53 have been developed and are under various stages of clinical trials. These include AMG-232, APG-115, BI-907828, CGM097, RG7388, DS-3032b and HDM201 [346,347,348].

### 4.3. Cancer Therapies Targeting Bcl-2 Family Proteins for Degradation

Several compounds that target various Bcl-2 family members have been developed. Of notable success is the Bcl-2-specific inhibitor Venetoclax (ABT-199), which has been FDA-approved for the treatment of chronic lymphocytic leukemia (CLL), small lymphocytic lymphoma (SLL), and acute myeloid leukemia (AML) [349,350,351,352,353].

More recently, intense efforts have been made to develop drugs that mediate protein degradation, rather than just binding and inhibiting target protein activity. A major advantage of this approach is the ability to target what were classically considered “non-druggable proteins” [354,355,356,357]. Proteolysis Targeting Chimeras (PROTACs) are small molecules that can promote the ubiquitylation of target proteins by directing specific E3 ligases to specific substrates [318,358,359]. Mechanistically, PROTACs form a ternary complex by binding the protein of interest and an E3 ligase, resulting in the ubiquitylation and degradation of the target protein [359]. For example, Bromodomain Extra-Terminal chimeric molecules (BET-PROTACs), such as ARV-825 and ARV-771, are capable of binding specific target proteins and inducing their ubiquitylation and degradation [360]. Recently, Bcl-xL, Mcl-1 and Bcl-2 PROTACs have been developed [361,362,363,364,365]. DT2216 is the most promising Bcl-xL-specific PROTAC and brings Bcl-xL to the Von Hippel-Lindau (VHL) E3 ligase for degradation by the proteasome [362]. DT2216 is derived from the ABT-263 and showed higher selectivity to kill Bcl-xL-dependent cancer cells than ABT-263 [361,362]. The dMcl1-2 and C3 PROTACs induce Mcl-1 degradation by bringing it into close proximity to cereblon (CRBN) cullin-4A RING E3 ligases [364,365]. In addition, C5 PROTAC was shown to potently and selectively induce the ubiquitylation and proteasomal degradation of Bcl-2 [364].

### 4.4. Cancer Therapies Targeting IAPs for Degradation

IAPs are overexpressed in various tumors, making them attractive drug targets for cancer therapy [366,367,368]. In the past few years, efforts have been made to target IAPs, and specifically XIAP, by small molecules “Smac-mimetics” (SMs) [280,281,369,370,371,372,373]. SMs are small molecules that were based on the evolutionary conserved tetra-peptide IAP-Binding Motif (IBM, AVPI/F). This motif was originally observed in the *Drosophila* IAP-antagonists Reaper, Hid, Grim and is also found in the mammalian IAP-antagonists Smac and Omi [270,271,273,274,280,370]. There are two classes of SMs: monovalent, which contain one AVPI binding motif and bivalent, which contain two AVPI binding motifs and are more potent than the monovalent [371,374]. SMs were originally designed to target and inhibit XIAP [269,372,375,376,377]. Although SMs can bind XIAP, they are not very effective in degrading it [247,373,374,378]. On the other hand, SMs efficiently promote the degradation of cIAPs via the UPS [283,379,380]. The SM-mediated inhibition of cIAPs causes apoptosis through the inhibition of NF-κB signaling [250,251,380,381,382,383,384,385,386,387]. SM130 and SM114 primarily target cIAPs for degradation but have reduced affinity towards XIAP [373]. TL32711 (Birinapant) is a bivalent molecule that works particularly well against cIAP1 and is well tolerated at doses that sustain target inhibition [388,389,390,391]. Unfortunately, most cancer cell lines are resistant to SMs [383,392,393]. Therefore, combinations with other anti-cancer drugs are being explored in an effort to overcome resistance [394,395,396]. Historically, SMs were developed to target XIAP, but until recently, no potent XIAP-only inhibitors were available [34,356]. Mamriev et al. reported of small molecule ARTS-mimetics that can bind XIAP and promote its degradation via the UPS [397]. This compound was identified in a virtual screen for small molecules with the highest docking affinity to the specific and unique binding site of ARTS within the BIR3 domain of XIAP [397]. These small-molecule ARTS-mimetics can induce apoptosis in a wide range of cancer cells but not in healthy PBMC (Peripheral Blood Mononuclear Cells). ARTS-mimetic small molecules bind specifically to XIAP, but not cIAP1, and promote the degradation of both XIAP and Bcl-2 through the UPS [397]. ARTS-mimetics act as PROTACs by bringing the E3 ligase XIAP to its target Bcl-2, thereby inducing the degradation of both these proteins [397]. ARTS-mimetics provide a promising novel platform for developing highly specific and potent anti-cancer drugs by targeting XIAP-and Bcl-2 for degradation [397].

Another strategy to target IAPs for degradation is by a series of chimeric molecules termed specific and non-genetic inhibitor of apoptosis protein (IAP)-dependent protein erasers (SNIPERs). SNIPERs consist of three distinct parts: a target protein ligand, an E3 ligase ligand and a linker between them [398,399,400,401]. They were shown to recruit IAPs and promote their targeted protein degradation [398,399,402,403]. Unlike the traditional PROTACs, SNIPERs induce simultaneous degradation of IAPs, such as cIAP1 and XIAP, along with their target proteins [399,404]. Although PROTACs and SNIPERs exhibited promising results in degrading target proteins, there are still some challenges to overcome before these compounds can be used in the clinic [401]. For example, most PROTACs do not obey Lipinski’s rule of five (RO5) because of their relatively high molecular weight [405]. In addition, PROTAC’s toxicity, bioavailability, distribution and metabolism still need to be determined [401]. For now, two PROTACs have entered phase I/II clinical trials— the PROTAC ARV-110 for the treatment of prostate cancer and ARV-741 for the treatment of breast cancer [401].

## 5. Future Directions and Challenges

The UPS is considered to be a major target for developing novel types of anti-cancer drugs. This was initiated with the use of Proteasome inhibitors, which were proven to be effective for patients with hematological malignancies (Mantle cell lymphoma and multiple myeloma) [406]. However, the efficiency of proteasome inhibitors is seriously compromised due to innate and acquired drug resistance [407,408]. Numerous studies have helped uncover the pathways responsible for drug resistance, making it easier to predict which patients can benefit from specific proteasome inhibitor therapy [409,410,411,412]. One of the approaches proposed to overcome drug resistance is combination therapy [406,412]. This might also help with treating malignancies, which present a limited response to proteasome inhibitors. In many cases, upregulation of E3 ligases is responsible for drug resistance [58,192,413]. Fortunately, many E3 ligases, such as cIAP, XIAP, MDM2 and others, have become popular targets for drug development, including specific small molecules, targeting E3-ligases. Nevertheless, there are still challenges to overcome, including the vast diversity of E3 ligases and the fact that E3 ligases can have various different substrates, including tumor suppressors and oncogenes [414]. The combination of small molecules with other cancer therapies showed better efficacy than monotherapies [414]. Besides small molecules, the protein-targeting chimeras (PROTACs) were developed to overcome drug resistance via hijacking the UPS mechanism. While PROTACs are showing promise in providing a novel approach to overcome drug resistance, many challenges await to be resolved regarding their drug design and possible clinical applications.

In summary, the UPS plays a major role in regulating key apoptotic proteins. As many as 20 E3 ligases alone are known to control the levels of the p53. This illustrates the importance of regulated protein degradation and governing the activity of this major tumor-suppressor protein. High levels of Bcl-2, XIAP and cIAPs are characteristic of many types of cancers and hence make these proteins attractive drug targets [366,367,368,415,416]. Interestingly, far fewer E3 ligases control the levels of these proteins compared to p53. A possible reason for this difference is that these apoptosis-suppressing proteins are regulated by direct binding to neutralizing proteins (such as Bax in the case of Bcl-2) or IAP-antagonists [35,263,417]. Perhaps this regulation through protein–protein interactions can complement any possible limitations resulting from the relatively small number of specific E3 ligases dedicated to these proteins. Degrading the target protein rather than binding and blocking their function has significant advantages; mainly, it reduces the load of the elevated expression of the target protein, which are often inhibitory proteins. Moreover, it can reduce systemic drug concentrations and, hence, possible cytotoxic side effects. Recently, major efforts have been devoted both in academia and by pharma companies to develop therapies that recruit the UPS for promoting apoptosis in cancer cells. Compounds that specifically target proteins for degradation resulting in effective tumor killing may dramatically improve the success of cancer therapy.

## Figures and Tables

**Figure 1 cells-10-03465-f001:**
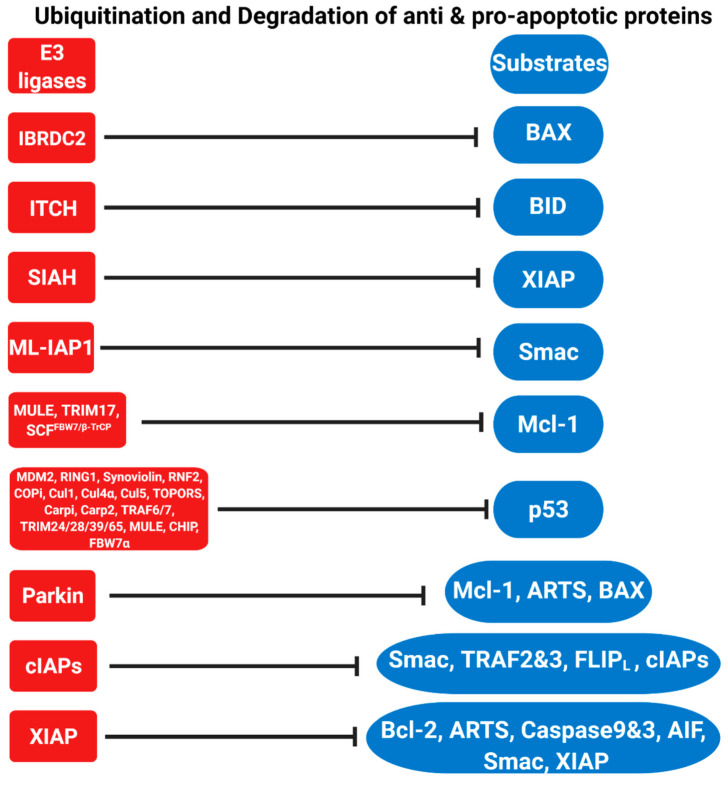
E3 ligases (red) regulate the apoptotic pathway by inducing the degradation of pro- and anti-apoptotic proteins (substrates in blue) via the ubiquitin proteasome system (UPS).

**Figure 2 cells-10-03465-f002:**
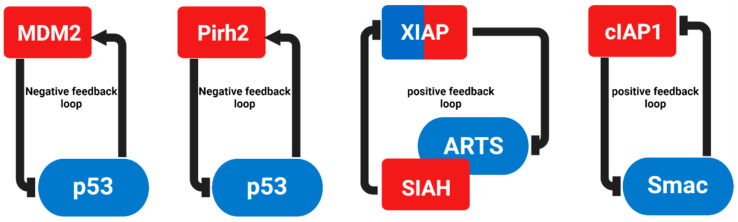
Each of these feedback loops consists of proteins whose levels and function are influenced by the activation or inhibition of their E3 ligase. Arrows indicate stimulatory interactions, whereas horizontal bars denote inhibitory influences.

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
