# Peer review of "Killing by Degradation: Regulation of Apoptosis by the Ubiquitin-Proteasome-System"

_cells, 2021, doi:10.3390/cells10123465_

Round 1
Reviewer 1 Report
This review manuscript is well written and comprehensive. The topic would have wide range of readership. The literature is up to date.
Minor,
Some editorial could improve the manuscript.
1, in tile, the use of (;) may be changed to (:).
2, in line 50, if it is not a journal style, then “ubiquitin proteasome system (UPS)” may be used, instead of “Ubiquitin Proteasome System”, and keep consistent throughout manuscript.
3, in line 72, the word, E3-ligase, may be replaced with “E3 ligase”, and keep consistent.
4, in line 92, “ligases-act”, may be “ligases act”.
5, in line p94, “protein” may be “proteins”.
6, in line 102, the UPS may be used.
7, in line 112, p53 may be italic, if used to describe a gene.
8, in line 211, E3-ligase (MULE/ARF-BP1) sentence may be modified.
……
9, the used pf articles “a” and “the” needs to be checked throughout manuscript.
Author Response
We want to thank the reviewer for his/her comments

Reviewer 2 Report
The review entitled ‘Killing by Degradation; Regulation of Apoptosis by the Ubiquitin-Proteasome –System’ is a comprehensive summary of the role of the UPS in regulation of apoptosis. The authors detail the different ubiquitin ligases that are involved in the degradation of pro and anti-apoptotic proteins with extensive focus on p53 and the Bcl-2 family of proteins. They further describe the current approach of targeting the UPS for apoptosis-induced cancer therapy. This article provides a very detailed account of the existing knowledge about the UPS and its function in regulating apoptosis.
Minor point:
The authors should discuss the future direction of such therapies, obstacles in new drug discovery, and challenges faced in getting drugs approved.
Author Response
thank you for you comment, a new paragraph was added to the manuscript as requested
